# A Literature Review of Modeling Approaches Applied to Data Collected in Automatic Milking Systems

**DOI:** 10.3390/ani13121916

**Published:** 2023-06-08

**Authors:** Laura Ozella, Karina Brotto Rebuli, Claudio Forte, Mario Giacobini

**Affiliations:** Department of Veterinary Sciences, University of Turin, 10095 Grugliasco, TO, Italy; laura.ozella@unito.it (L.O.); karina.brottorebuli@unito.it (K.B.R.); mario.giacobini@unito.it (M.G.)

**Keywords:** dairy cows, automatic milking system, algorithms, modeling approaches, statistical analyses, machine learning, mastitis detection, milk production, cows’ behavior

## Abstract

**Simple Summary:**

Automatic milking systems (AMSs) are revolutionizing dairy farming worldwide. Not only do they control the milking process, but they also bring changes to the whole farm system management. In this review, the examination focused on the study of AMSs using various modeling approaches. Our review primarily encompassed published articles addressing cows’ health, production, and behavior/management. Within this field, Machine Learning (ML) emerged as the prevailing modeling approach. Most of the studies were aimed at detecting cows’ health problems, especially mastitis. However, there is still a lack of a robust methodology for utilizing ML techniques in this domain, and it was also observed that the number of positive and negative cases is often unequal, leading to populations that are not balanced when predicting health issues. Only a small number of studies focused on milk production, even though accurate forecasting of individual cow milk yields could be very useful. Additionally, the study of cow behavior and herd management using AMSs is still not very explored.

**Abstract:**

Automatic milking systems (AMS) have played a pioneering role in the advancement of Precision Livestock Farming, revolutionizing the dairy farming industry on a global scale. This review specifically targets papers that focus on the use of modeling approaches within the context of AMS. We conducted a thorough review of 60 articles that specifically address the topics of cows’ health, production, and behavior/management Machine Learning (ML) emerged as the most widely used method, being present in 63% of the studies, followed by statistical analysis (14%), fuzzy algorithms (9%), deterministic models (7%), and detection algorithms (7%). A significant majority of the reviewed studies (82%) primarily focused on the detection of cows’ health, with a specific emphasis on mastitis, while only 11% evaluated milk production. Accurate forecasting of dairy cow milk yield and understanding the deviation between expected and observed milk yields of individual cows can offer significant benefits in dairy cow management. Likewise, the study of cows’ behavior and herd management in AMSs is under-explored (7%). Despite the growing utilization of machine learning (ML) techniques in the field of dairy cow management, there remains a lack of a robust methodology for their application. Specifically, we found a substantial disparity in adequately balancing the positive and negative classes within health prediction models.

## 1. Introduction

The introduction of automated milking systems (AMSs), or milking robots, in the early 1990s, represented one of the major headways in dairy farming techniques. Automatic milking is based on cows’ voluntary visits to the robot so that cows are no longer brought to the milking parlor two or three times daily by human handlers. Animals are free to go to milking at any time on a daily basis as well as to dynamically change intervals between milking throughout the lactation period [1]. The AMS process is entirely mechanized, reducing the labor burden on farmers in relation to milking operations. This has the potential to enhance the quality of their work and improve their overall lifestyle. AMS has the potential to increase milk production in cows as they can be milked up to three times a day on average, compared to twice-daily milking in conventional systems. Studies have shown an increase in milk production ranging from 3% to 25% with the use of AMS [2]. The increased frequency of the number of milkings also reduces the external udder pressure when lying whilst at the same time reducing stress on the udder ligaments, thus increasing the comfort of the animal [3].

During each milking, automatic sensors allow the monitoring of udder health and milk quality by providing detailed information about each cow, which was not easily obtained with previous conventional systems [4]. Moreover, the cows may benefit from the freedom to control their physical activity and to reduce potential stress at the time of milking avoiding gathering and crowding phenomena usually present in conventional parlors [5]. Therefore, the adoption of AMS has grown significantly worldwide with an estimated 50,000 units on 25,000 farms in operation in 2019 [6]. The implementation of AMS technology not only facilitates the collection of milk quantity and quality data but also creates an opportunity to study cow behavior and welfare within a system that allows free cow traffic. The introduction of AMS has altered the daily rhythms and behavior of cows throughout their lactation cycle, making it important to consider both AMS efficiency and animal welfare. Despite the large amount of data collected in AMS-equipped farms, there is still a significant potential for herd characterization and management optimization that remains largely unexplored [7].

However, the impressive increase in newly available data, precious for researchers, could become too complex for farmers, running the risk of having little feedback in daily farm management. This is included by the phenomenon of “Big Data”, i.e., massive volumes of data with a wide variety that can be analyzed and used for decision-making [8]. Improving data integration is crucial to fully utilize the vast amount of data and make the resulting information easily accessible to farmers [9]. However, the complex and nonlinear relationships hidden within large and often redundant data are hard to unravel using traditional statistical models [10]. Machine Learning (ML) is a subfield of Artificial Intelligence that uses complex algorithms and complementary data modeling techniques to those used in traditional statistics [11]. One of the key benefits of using ML techniques is that they can effectively identify meaningful relationships within large, complex, and often redundant datasets from multiple sources. In general, ML methods involve a learning process where the model learns to perform a task by analyzing and processing a set of training data. Over time, the performance of the ML model is continuously improved by refining the model’s learning algorithm to better analyze and interpret the data. ML approaches are often referred to as data-driven since the algorithms rely on learning from the data. They can provide more accurate results than traditional statistical approaches, which may be influenced by the researcher’s preconceptions or hypotheses [9]. Recently, ML techniques have been applied to Precision Livestock Farming (PLF) [12], with applications in livestock management and productivity, e.g., [13,14], animal behavior and welfare [15,16]. In the dairy sector, ML methods are being used for various tasks, such as estrus detection [13,17], heat stress severity evaluation [18,19], and social interaction tracking [20]. Although AMS have become increasingly popular in dairy farming, there remains a lack of understanding regarding the specific algorithms used, the challenges faced, and the problems addressed by applying ML techniques to the data collected from the milking robot sensors and other sensors in AMS-equipped farms.

Motivated by the rapid advancements of ML, its increasing global popularity, and its potential influence on PLF (Precision Livestock Farming), we conducted a literature review focusing on various modeling approaches, including ML, using data from farms equipped with AMS for the analysis of animal health, production, behavior, and management. It is anticipated that the dairy farm sector will continue to see the increasing adoption of ML in the future, and the results of this review will guide and help the researchers and practitioners on how the adoption of ML could support the monitoring of dairy cows in AMS. The remainder of this review is structured as follows. The Section 2 introduces the review methodology, including the database and keywords used in literature retrieval, as well as the search results. The Section 3 briefly describes the most widely used ML techniques and their main performance metrics. The Section 4 shows an overview of the works related to modeling and AMS in the field of health, behavior, and production. In the Section 5, the review concludes with future research directions according to our analysis of the previous studies.

## 2. Review Methodology and Results

The articles included in this review were primarily sourced from the Web of Science and Google Scholar platforms. To identify relevant scientific articles, we utilized keywords encompassing various categories, such as “Machine Learning”, “Artificial Intelligence”, “modeling approaches”, “Automatic Milking Systems” (or “milking robots”), and “Precision Livestock Farming” (or “Precision Dairy Farming”), both abbreviations and full names. In addition to conducting keyword searches, we also carefully examined the cited references in the published literature, ensuring that these articles align with the scope of our search. The published time of the selected literature in this work was restricted to the last 22 years, i.e., from 2000 to 2022. Finally, we selected and thoroughly reviewed 60 publications from the retrieved results. In addition, we also consulted other relevant articles and supporting literature, including other reviews, to provide comprehensive insights. Given the wide range of applications of ML in Precision Livestock Farming (PLF), numerous reviews have been published within this research field, e.g., [12] and in particular, on dairy farming [21,22]. The selected 60 studies were classified into three generic categories: health, production, and behavior/management. Most of the studies were intended for the detection of cows’ health problems (82%), in particular, mastitis; 7% of the papers focused on cows’ behavior/management and 11% focused on milk production.

The 60 articles included in the review are summarized in Appendix A), which provides information on the application domain, the addressed problems, the modeling techniques used in the analyses, the datasets, and the list of variables used. Figure 1 illustrates the geographical distribution of the contributing studies in modeling approaches in farms equipped with AMS, considering the location of the dataset collection. It is noteworthy that investigations into modeling approaches in AMS are not distributed worldwide, with most studies originating from European countries (71%). This reflects the distribution of milking robots in farms, which are not yet widely present in developing countries.

Figure 2 displays the temporal trend of the number of publications from 2000 to 2022. The number of publications showed a peak in 2010, and it increased slowly in the last few years, starting from 2015.

## 3. Machine Learning Algorithms and Performance Metrics

ML, as a subfield of Artificial Intelligence, empowers computers to learn from data without the need for explicit programming [23]. It comprises a diverse set of algorithms, each with distinct objectives and learning strategies. According to the type of learning, ML methods can be grouped into three primary categories: supervised, unsupervised, and reinforcement.

In supervised learning, a predictive model is developed using the labeled data with the prior knowledge of the input and the desired output [24]. The goal of the supervised learning approach is to produce an inferred function that can be utilized for mapping new observations based on a set of training examples [25,26]. Thus, supervised ML algorithms are those which need external assistance. The input and the output are known, and the algorithms aim to discover the optimal approach to achieve the output based on the input. This process is divided into two phases. The first phase consists of the training phase where a collection of data samples is used to build or improve a computer model by learning from inherent structure and relationships within the data. The second phase consists of applying this computer model to new unseen observations to predict certain properties of these new samples. The generalization ability of the algorithm is monitored by partitioning the data. The dataset is split into two sets: the training and the test sets. The latter is not seen by the algorithm during the learning phase. It is used to assess the performance of the algorithm on unseen data, which gives an assessment of the generalization ability of the algorithm.

Unsupervised learning generally involves the analysis of unlabeled data under assumptions about structural properties of the data without prior knowledge of the input and output variables [24,27] These are called unsupervised learning because, unlike supervised learning, no labels are provided, and the algorithms discover and present the structure in the data.

Reinforcement learning refers to the challenge encountered by an agent as it learns behaviors through iterative interactions with a dynamic environment, relying on trial and error [28]. The information available in the training data provides an indication as to whether an action is correct or not [27], instead of indicating the correct output for a given input. Reinforcement learning algorithms are used, for example, for real-time decision-making and robot navigation [24]. Figure 3 provides an overview of the different types of supervised and unsupervised ML problems categorized by the data type (discrete or continuous) and grouped into four main classes: classification, clustering, regression, and dimensionality reduction based on their learning styles.

The main difference between ML and traditional statistical data analysis and deterministic modeling is that statistical and deterministic models work under the assumption of a hypothesis, mathematically represented by a model, to be analyzed or fitted based on the data. In ML, the algorithm itself is in charge of finding this model [29]. Simply put, statistical and deterministic analysis is model-driven. The former is based on a model-family that should be fitted to define the specific model instance that represents the data. The latter is based on a dynamic model that mathematically represents the rules that operate in the system, and the mathematical behavior of this model is studied. On the other hand, ML is data-driven, the algorithm rules are set to find the specific model that better fits the training data, but this model does not have any predefined form. However, in the literature, this distinction is not always considered, and many articles consider statistical techniques as ML analysis.

In this review, we have included articles that used deterministic models, statistical analysis, and both supervised and unsupervised ML algorithms, while no work selected used reinforcement learning. Table 1 provides a summary of prominent supervised and unsupervised modeling techniques mentioned in this review.

Performance metrics are essential for evaluating the effectiveness of a model, but the choice of metric can significantly impact the importance placed on different aspects of a model’s performance. Therefore, selecting an appropriate performance metric is crucial for accurately assessing modeling performance [40]. In the context of classification algorithms, the confusion matrix constitutes one of the most intuitive metrics for finding the correctness of a model. For binary classification modeling, the confusion matrix is a 2 × 2 table having two dimensions, namely “Actual” and “Predicted”, and its values are the outcomes of the comparison between the predictions with the actual class label (Figure 4).

True Positives (TP) represent the total number of data points that have been correctly predicted as positive values and True Negative (TN) represents the total number of data points that have been correctly predicted as negative values. On the other hand, False Positives (FP) represent the total number of data points that have been predicted as positive values when they were actually negative, while False Negatives (FN) represent the total number of data points that have been predicted as negative values when they were actually positive. The aforementioned values can be implemented to estimate several performance metrics (Table 2).

In addition to these measures, the Receiver Operating Characteristic (ROC) curve is also used to evaluate the performance of a classifier. Figure 5 gives an example of this curve. It is a plot of the TP rate versus the FP rate.

Each point on this curve is the combination of TP and FP rates for a given value of the threshold used by the classifier to separate the classes. A robust classifier is expected to not increase the FP rate if the TP rate increases and vice-versa. On the other hand, a random classifier will directly increase the FP rate with the increase in the TP rate, as indicated by the dashed gray line in the plot. The ROC curve visually helps in analyzing the balance between these rates, since how close to the top-left corner is the plot, the more robust is the classifier. The Area Under the ROC (AUR) curve gives a value of the quality of the classifier and it is especially useful when comparing two or more classifiers.

## 4. Application of Modeling Approaches in AMS

### 4.1. Health

#### 4.1.1. Mastitis

Modeling tools have been widely used in predicting mastitis based on data from AMS, proving useful in addressing the economic losses and welfare concerns associated with this disease in the dairy industry. Mastitis is a highly concerning condition that results in decreased milk production, reduced milk quality, and compromised cow welfare [41]. Detecting mastitis in its early stages is crucial for improving both milk production and cow welfare. This disease can present in clinical or subclinical forms, with the latter being up to 40 times more common than the former [42]. Subclinical mastitis is more difficult to detect than its clinical form as clinical signs are not evident in the infected cow. Subclinical mastitis may go unnoticed due to the absence of visible signs of inflammation or significant changes in milk composition. Therefore, the timely detection of subclinical mastitis is crucial in order to implement appropriate treatment, control, and preventive measures. With the introduction of AMS in farms, the identification of mastitis is no longer established through direct visual observation, and the control of the health status is based on sensor measurements [43]. Several studies evaluated the performance of automated mastitis detection systems with respect to their practical value for farmers and investigated the specificity and sensitivity of the system. Aspects of milking machine design and performance are addressed in standards issued by the International Standards Organization (ISO). The International Standard ISO/FDIS 20,966 describes a minimum sensitivity of 80% combined with a specificity higher than 99% as a requirement for a reliable mastitis detection system [44], but these recommendations are still under discussion.

##### Variables Used to Detect Mastitis

Milking robots can collect various data during milking, including milking time, milk yield, Electrical Conductivity (EC), Somatic Cell Count (SCC), and milk components. Therefore, there is an opportunity to integrate these measurements into disease detection models for mastitis. Identifying the most effective variables for the detection of clinical and subclinical mastitis is of paramount importance. SCC, which reflects the inflammatory status of the mammary gland, is historically the most used predictor of mastitis, and it is an indicator of both resistance and susceptibility of cows to mastitis [45]. A threshold of 200,000 cells/mL of SCC is commonly used to identify subclinical mastitis [46]. However, in cases where the SCC is very low (below 50,000 cells/mL), the Differential Somatic Cell Count (DSCC), which considers the ratio of neutrophils and lymphocytes, becomes a valuable tool for enhancing the identification of the mammary gland status in dairy cows [47]. At the same time, the SCC day test used for mastitis surveillance, gives data that fluctuate widely between days, creating doubts of its reliability [48].

Therefore, the utilization of a combination of different indirect indicators of mastitis could potentially yield more accurate results in the detection of the disease. Other detection systems are based on testing the EC of milk. EC is an indicator of ionic changes in the milk that occur during mastitis. Norberg et al. [49] noted that the electrical conductivity (EC) of milk could serve as a significant trait for mastitis detection, as cows affected by the disease often exhibit higher conductivity in their milk. These authors applied a Generalized Linear Model to distinguish between healthy and unhealthy cows based on EC. Their results indicated that cows with mastitis may not always show an increased EC; nevertheless, the variation in EC from infected quarters may be greater than the variation in EC from healthy quarters. The results of this study showed that EC has poor diagnostic test properties for the detection of subclinical mastitis. In fact, the accuracy for clinical cases was 80.6%, while the accuracy for subclinical cases reached only 45% [49]. Later studies demonstrated that the combination of EC and SCC improved the detection of subclinical mastitis in detection algorithms [50,51]. Using EC and SCC information, a Fuzzy Logic algorithm produced a two- to three-fold increase in the success rate (i.e., positive predictive value) and a two- to three-fold decrease in the false alert rate, compared to the use of EC alone [50]. Additionally, quarter-level (i.e., every quarter of each cow was considered as an independent unit) SCC assessment results in higher sensitivity and specificity than cow-level assessment, even when combined with EC measurement [50,51]. In agreement with these studies, [52] also achieved poor mastitis detection results with the use of EC alone. The authors tested four different detection methods, namely: threshold of EC, creation of indexes ad hoc and control chart of cumulative sum applied directly to EC data and to the residuals of a Linear Mixed Model with lactation and parity as fixed effects and the cow and quarter as random effects. Since the model was unable to achieve the ISO standard sensitivity (>80%) and specificity (>99%), the authors suggested that improvements can be achieved by using other parameters, such as milk yield, milk flow, and composition analysis, to increase the method’s accuracy and thereby improving the utility of mastitis detection systems. Sun et al. [53] used the combination of EC and quarter milk yield to detect clinical mastitis, by the application of two types of Neural Networks (NN), Multilayer Perceptron (MLP) and Self-Organizing Map (SOM). The MLP model achieved 91% of sensitivity and 87% of specificity when using data transformed by Principal Component Analysis. The SOM, using K-means as a clustering algorithm, revealed three clusters that reflected the stage of progression of mastitis in a quarter: healthy, moderately ill, and severely ill. Other parameters such as milk color and milk yield are associated with SCC and EC to classify abnormal milk, often caused by clinical mastitis [42]. In a study by Hovinen et al. [54] milk color was used for the detection of mastitis in addition to the EC. Their findings showed that the specificity for EC was quite high, but the false alert rate was also high. In 11 out of 17 cases, clinical mastitis was detected during a 6-days period before clinical signs were detected. Five of those were detected solely based on milk color and not on EC. We can conclude that milk color added value to the detection system. In agreement with these results, Kamphuis et al. [55] suggested that mastitis detection performance might be improved by combining different predictive variable types, including milk colors and milk production. They found that green and blue colors were the best indicators for both abnormal milk and clinical mastitis [55]. However, in a study conducted by Altay et al. [56] on two breeds of dairy cows (Holstein and Brown Swiss), a Logistic Regression model showed that only SCC and EC were effective variables on mastitis detection, but not other variables, including the milk color.

Alternative markers, such as Lactate Dehydrogenase (LDH), have been suggested as potential indicators for early mastitis detection and diagnosis [57,58,59]. LDH is an enzyme present in the cytoplasm of all cells in the body, and during an inflammatory process such as mastitis, it is released from the cells into the milk due to cell damage [60]. In dairy milk, LDH is correlated with SCC [61]. Chagunda et al. [57] developed a dynamic deterministic model using LDH as the main indicator measured in milk and obtained sensitivity and specificity for detecting clinical mastitis at a level of 82% and 99%, respectively. In their study, healthy cows were defined as having no veterinary treatment within the incurrent lactation period and an SCC <100,000 cells/mL, and the additional factors incorporated in the model are days from calving, breed, parity, milk yield, udder characteristics, other disease records, EC, and herd characteristics. As biosensor assays for enzymes like LDH in milk are now becoming available, they provide an opportunity for automated, real-time mastitis detection. Friggens et al. [58] successfully tested the mastitis risk model by Chagunda et al. [57] for individual cows based on LDH on a scale from 0 (completely healthy) to 1 (full-blown mastitis) for the early identification of acute mastitis cases (4 days before treatment). Ankinakatte et al. [59] also used LDH in addition to SCC and EC as indicators and evaluated the performance of NNs and Generalized Additive Models to predict mastitis. The study showed similar performance for the two models, even though the inclusion of SCC improved their predictive ability by >5%, thus confirming the importance of this parameter in the detection of mastitis. Penry et al. [62] proposed that quarter peak milk flow rate was the variable associated with increased risk of clinical mastitis as a primary hypothesis. They conducted a retrospective, case-control study using a Logistic Regression model, and included in the model five predictor variables besides the peak milk flow rate, i.e., parity, quarter position, day in milk at diagnosis of clinical mastitis, udder milk yield, and milking interval. However, only the milking interval, but not the quarter peak milk flow rate was associated with the risk of clinical mastitis.

More recently, Naqvi et al. [63,64] developed a Recurrent Neural Network model for the detection of clinical mastitis by comparing numerous subsets of variables to determine their importance and impact on model performance. They integrated several variables that are regularly measured on AMS farms (including milk and behavioral characteristics, cow traits and farm-level/environmental variables) but have typically been excluded from mastitis detection models. Their results showed that SCC, the variance in the milk intervals observed during the day, and milk temperature were identified as the three most significant variables based on their impact on the model predictions. Interestingly, eight of the top twenty variables were behavioral measurements (such as activity, rumination, and milking duration), suggesting they can play a role in the detection of mastitis. The significance of SCC as a crucial mastitis indicator was showcased in a study by Bonestroo et al. [65], which designed a prediction model utilizing gradient-boosting trees to detect subclinical mastitis. The model accurately predicted whether SCC would decrease below the 200,000 SCC/mL threshold within 50 days after an initial increase in SCC, using 30 days of sensor data. However, altering the input requirement from 30 days to 15 days had a negligible impact on the performance of the model.

##### Mastitis Alert List

Dairy farmers using an AMS often complain about the high number of false positive (FP) alerts on the mastitis alert lists. These alerts can lead to an overestimation of the number of animals diagnosed and treated for mastitis, which is a concern for both animal welfare and production losses. Therefore, reducing the number of FP alerts is crucial for improving the specificity of the system, particularly when milk is automatically separated. As suggested by Mollenhorst et al. [66], an ideal monitoring system would produce a low number of false alerts while alerting with emphasis on the more severe cases and in a timely manner (a maximum of 24 h before the onset of the disease as desirable). A detection model that includes Time Series regression models for two variables (milk yield and EC) was used to generate mastitis alerts [67]. In this study, the model outcomes (mastitis alerts) were compared to actual occurrences of clinical mastitis. A case of mastitis was classified as a true positive (TP) if one or more alerts were given within a defined period around the recorded date of an observed case. Otherwise, the case was classified as a false negative (FN). Applying the Time Series models led to a better performance of AMS in mastitis detection by the reduction in the number of FP alerts. Similarly, the application of Fuzzy Logic gave an important improvement in decreasing the number of FP test results [68]. With a Fuzzy Logic model for classifying alerts of clinical mastitis as true or false, the number of FP alerts could be reduced. De Mol and Woldt [68] detected which animal was infected with mastitis based on traits for each cow and the model achieved a sensitivity of 100% and a specificity higher than 99%, the FP was reduced by 95% while the number of TP alerts remained at the same level [68]. This improvement is certainly beneficial for herd management support. More recently, Khamaysa Hajaya et al. [69] proposed a NN to build a mastitis detection model. The model achieved a specificity of 99%, and sensitivity of 97%, demonstrating that, with this high specificity, and relatively high sensitivity, the model could reduce the problem of FP alerts.

Combining the probability of clinical mastitis based on AMS sensor measurements with non-AMS cow information was proposed as a way to improve disease detection [70]. The authors presented a method in which a previous probability of clinical mastitis (based on parity, DIM, season, SCC history and clinical mastitis history, i.e., throughout the cow’s lifetime) was combined with the test characteristics (sensitivity and specificity) of the AMS detection system to discriminate between alerts. A Tree-augmented Naïve Bayesian Network was trained using available data to calculate cow-specific prior probabilities for clinical mastitis. Results showed a similar specificity and sensitivity of the system when different types of information were included or with the AMS alerts only. Moreover, in this study, the additional value of non-AMS cow information to discriminate between TP alerts and FP alerts was not specifically investigated. Steeneveld et al. [44] used a Naive Bayesian Network as a method (70% sensitivity and 97.8% specificity) for discriminating between TP and FP alarms in the detection of clinical mastitis. This study reported a minor effect of using non-AMS cow information on making a distinction between TP and FP mastitis alerts. Thus, according to the authors, the use of additional non-AMS data did not add much to the detection performance of sensor systems. Nevertheless, the effect of a combination of AMS sensor data and other cow information on FP needs to be further investigated. Kamphuis et al. [71,72] used EC, milk production, dead milking time, and milk flow with a Random Forest (RF) algorithm for clinical mastitis detection. RF are ensembles of Decision Trees (DT), which were created with Bagging and Boosting. Bagging consists in creating different DT for different Bootstrap samples of the dataset. The final model outcome was the average of all models. Boosting consists in creating different DT classifiers sequentially in such a way that the next model gives more weight to the instances that were incorrectly classified by the previous classifier. Both studies achieved a high specificity but a low sensitivity for clinical mastitis, using a narrow timeframe, and they concluded that RF made it possible to decrease the number of FP alerts by more than 50%. However, extending the length of the time window greatly enhances the apparent sensitivity and specificity of detection systems. The authors demonstrated that increasing the length of the time window significantly affects performance indicators: using a 24-h time window preceding the occurrence of a clinical mastitis episode resulted in a sensitivity of 40% at a specificity of 99%. Increasing the time window to 96-h preceding the occurrence until 72-h after the occurrence of a clinical mastitis episode increased sensitivity to 75% at the same specificity level of 99% [72]. Bausewein et al. [73] recently identified parameters that could enhance the sensitivity and specificity of AMS alerts when analyzed by farmers after each milking. The study also revealed minor variations in mastitis alerts among manufacturers, likely attributable to differences in sensor technology and proprietary algorithms.

##### Mastitis Indicators

Mastitis infection has traditionally been viewed as dichotomous (healthy vs. sick), which is convenient for clinical treatment and assessing treatment efficacy based on clinical symptoms. However, this perspective does not capture the actual progression of the infection. It might be useful to get away from a binary mastitis variable and go to a continuous mastitis variable. Some authors introduced the measure of Degree of Infection (DOI), i.e., not a dichotomous quantity but a continuous varying quantity [58,74]. Friggens et al. [58] investigated the use of a dynamic deterministic model to detect the DOI on a scale ranging from 0 (completely healthy) to 1 (full-blown mastitis) for the early identification of acute mastitis cases. This was achieved by analyzing continuous data on SCC and comparing it to the levels of LDH in cow milk. Their model, mainly based on LDH measurements, was able to detect significant differences between cows with mastitis (mastitis risk 0.12) and healthy cows four days before treatment. Later, Højsgaard and Friggens [74] demonstrated that by combining a panel of measures reflecting different aspects of mastitis (EC, SCC, LDH) it is possible to derive a DOI measure that is a considerable improvement in precision relative to binary healthy/sick type classifications of health status. Sørensen et al. [75] proposed the use of an Elevated Mastitis Risk (EMR) indicator for detecting cases of clinical mastitis. This indicator is a continuous variable ranging from 0 to 1, where values closer to 0 indicate a low risk of mastitis, and higher values approaching 1 indicate an increased risk of clinical mastitis [75]. The estimated EMR values were utilized to generate two types of alerts: new infection alerts and ongoing intramammary infection alerts. The algorithm developed by the authors yielded a high specificity of 99%, but a low sensitivity: between 28% and 43% when reporting new mastitis cases, and between 55% and 89% when indicating ongoing intramammary infections.

##### Comparison between Modeling Approaches to Detect Mastitis

Comparing the performance of different modeling techniques for mastitis detection is challenging due to differences in mastitis definition and data properties. However, comparing studies that used different models on the same dataset is possible. Ideally, the studies should also use the same data partitioning, but this is not always the case. Therefore, a comparison of different models is presented here, highlighting the data partitioning used in each study. Cavero et al. [43,76] used a dataset of 403,537 milkings involving 478 cows to develop classification models for early mastitis detection, and in both studies, mastitis was determined according to udder treatments or SCC. Cavero et al. [43] developed a model that incorporated EC, milk yield, and milk flow rate. They applied a Fuzzy Logic classification model to aid decision-making that classified results as indicating mastitis, different degrees of likelihood for mastitis, or no mastitis. The authors trained the model with two thirds of the data and left the remaining third for test data. They evaluated the model according to sensitivity, specificity and error ratio and reported that the specificity of mastitis diagnosis changes between 75.8% and 93.9% and the error ratio varies from 41.9% to 95.5% when the sensitivity ratio is at least 80%. In the later research, Cavero et al. [76] constructed their classification system by application of NN using the following variables: EC, milk yield, milk flow and days in milk. Four different NN were used, trained with the backpropagation algorithm, and containing one neuron in the output layer (presence or absence of mastitis). The model was trained with a five-fold cross-validation data partitioning. Mastitis cases were correctly identified between 51.3% and 80.5%; however, the results were inferior in comparison with those obtained in the previous study. In particular, in [43] specificity and error rates obtained with Fuzzy Logic were found to be better compared to the estimates in Cavero et al. [76] through the use of NNs. The same results were obtained by Krieter et al. [77] that used this same dataset to investigate the usefulness of NN in the early detection and control of mastitis. The only difference to Cavero et al. [76] is that they used a different data partitioning, with 80% of data records for the training set and 20% for the test set without cross-validation. The specificity and error rate obtained with Fuzzy Logic [43] could be found to be better compared to the estimates obtained by Krieter et al. [77]. Mammadova and Keskin [78,79,80] detected the presence of subclinical mastitis by applying four different ML algorithms (NN, Adaptive Neuro Fuzzy Inference System, Fuzzy Logic, Support Vector Machine—SVM) on the same dataset. They used four different data partitions: 90%, 75%, 70% and 60% of the data for the training set and the remaining data for the test set. The overall result presented by the authors is the average of these four data partitioning solutions. Mastitis alerts were generated using input data such as lactation rank (current lactation number), milk yield, EC, average milking duration, and season. The SVM was the best model prediction of subclinical mastitis with a sensitivity of 89% and specificity of 92%.

Ankinakatte et al. [57] reported predictions obtained using different modeling methods on data collected at various time points. The authors evaluated the performance of NN and Generalized Additive Models (GAM) in terms of sensitivity and specificity. Similar results of 75% sensitivity and 80% specificity have been reported from NNs, and GAM, though their results indicated that the performance of the GAM model was slightly superior to that of the NNs, depending on the input variables utilized, and the inclusion of SCC improved the predictive ability of both models by >5%. Additionally, Anglart et al. [81] reported GAM to be a good predictive method to detect cow composite SCC, instead of predicting the mastitis events, by using quarter and cow milk data regularly recorded in cows milked in an AMS in an 8-week trial. The authors evaluated three modeling methods (GAM, Random Forest, and Multi-layer Perceptron—MLP), all with the five-fold cross-validation data partitioning and found GAM and MLP to be promising for udder health prediction. Ebrahimi et al. [82] applied several modeling techniques (NN, Naive Bayes, GLM, Logistic Regression, DT, Adaptive Boost—AdaBoost and Random Forest—RF) to determine the best model that could predict the risk of sub-clinical mastitis. Data from 364,249 milking instances were collected and milk volume, lactose concentration, EC, protein concentration, peak flow and milking time were analyzed using a 10-fold cross-validation. Overall, they found a high sensitivity (>93%) of all employed models, demonstrating the high distinguishing power of these models in the reliable identification of sub-clinical mastitis. However, the general low specificity showed a lower power to identify healthy samples of the tested models. The study concluded that the AdaBoost algorithm provided the best accuracy of 84.9% from the former parameters; however, the RF algorithms showed a similar level of accuracy (82.3%) [82]. Recently, a comparison study on ML methods was performed using data from both AMS and traditional milking parlors [47]. Eight different modeling methods (Linear Discriminant Analysis—LDA, Logistic Regression—LR, Naive Bayes, classification, and regression Decision Trees—DT, k-NN, SVM, RF and NN) were compared to predict subclinical mastitis based on SCC on a test set with 20% of the data observations. High specificity and the best precision were observed for SVM, LR and LDA. On the contrary, k-NN achieved the highest accuracy (>94%) compared to RF, SVM, and AdaBoost models in a study conducted by Tian et al. [83] to detect clinical mastitis. However, the small dataset used in this study (60 cows; 54 for the training set) may not represent population characteristics of conditions induced by mastitis and it does not allow generalizations on the performance of the algorithms used. Interestingly, an open-source ML application was recently developed to predict the risk of mastitis [84]. To achieve this goal, 26 classification models were built without any hyperparameter tuning and using 80% of the data for the training phase. The best-performing model proved to be the RF model and it was, then, tuned with 10-fold cross-validation. Its results had an accuracy of >98%, and sensitivity and specificity of 99.4% and 98.8%, respectively. The application could be integrated into AMS to detect the risk of mastitis in real-time.

##### Presence of Mastitis Pathogens

Models that utilize milk parameters to detect the presence of mastitis-causing pathogens can provide valuable information for managing the disease. After consulting the mastitis alert lists, farmers must be aware of the causal pathogen to initiate an effective antimicrobial treatment [85]. Bacteria that cause mastitis can be grouped into contagious or environmental, Gram-positive, or Gram-negative, or major and minor pathogens, according to their prevalence and the severity of symptoms. Hassan et al. [86] focused on using both unsupervised (USNN) and supervised (SNN) neural network models to detect small and large pathogens that cause bovine mastitis based on changes in milk parameters. They observed that SCC, protein percentage in milk, and EC showed to be the best predictors for major pathogen infections. SCC was also useful for differentiating minor pathogens of intra-mammary infections. Both USNN and SNN models were able to detect pathogen cases with a high degree of accuracy, with the USNN model providing a better overall result in terms of sensitivity (89% for minor pathogens, and 80% for major pathogens) and specificity (close to 99% for all bacteriological states). It was concluded that this model was better compatible with the results obtained from traditional microbiological methods. Bayesian Networks (BN) were employed to assess the likelihood of causal pathogens and associated risk factors. This approach utilizes conditional probability to uncover the connections between risk events and diseases [87], providing farmers with valuable insights for making informed management decisions, beyond mere pathogen identification [88]. Steeneveld et al. [88] classified mastitis bacteria as Gram-negative or Gram-positive using data that could be made available in AMS and cow information (e.g., parity, lactation stage, and history of clinical mastitis). The accuracy of classifying clinical mastitis cases into Gram-positive or Gram-negative pathogens obtained by applying naïve BNs was 73%. In a later study, a Decision Tree model was used to predict the Gram-status of clinical mastitis causal pathogens in conjunction with sensor information from the EC, milk color, and milk yield [85]. The authors failed to provide evidence to predict the Gram status of causal pathogens when EC values were used for only one day, suggesting that considering the temporal pattern of the EC may be beneficial. In addition, the results of the study suggest the potential of using milk color as a causal pathogen detection or prediction tool [85]. Castro et al. [89] described the distribution of mastitis pathogens in milk samples collected from several farms and identified the operational reliability and sensibility of mastitis alerts using a classification model. The average sensitivity and specificity of the mastitis detection system were 58.2% and 94.0%. Moreover, they found a high prevalence of environmental and contagious mastitis pathogens, due to the incomplete cleaning and disinfection of milk liners and teat dipping cups in AMS. The authors emphasized the importance of prioritizing prevention and control measures for pathogens, particularly in the context of all cows being milked with the same machine and using AMS, where milk cups are not disinfected between cows [89]. The use of commonly measured milk parameters in conjunction with ML techniques is a promising tool for detecting specific mastitis-causing pathogens and they should make detection systems more robust. However, given the conflicting results of the studies carried out so far, this potential still needs to be explored further.

#### 4.1.2. Other Diseases

Only a few studies explored the potentiality of modeling approaches for the detection of dairy cows’ diseases, besides mastitis, using data from farms equipped with AMS. Health problems are associated with reductions in activity, rumination, and milk yield. Therefore, the use of this information to detect a disease status is advised. Liberati and Zappavigna [90] employed a combination of milk production, milk flow, and animal activity measurements to detect abnormal cow health. They utilized a Fuzzy Logic model and Linear Discriminant Analysis (LDA) for this purpose. The reliability of these models in detecting various animal conditions, such as lameness, mastitis, and ovarian cysts, was assessed by comparing the generated alarms with the farm observations. Although both models showed limited accuracy in detecting specific abnormalities, the fuzzy model demonstrated efficiency in distinguishing between “normal” and “not normal” statuses. This capability is valuable for dairy herd management as it enables the identification of abnormal conditions before direct observation by the farmer [90]. Data associated with AMS and recorded by neck collar monitors (rumination and activity) can be combined to make management decisions more efficient, which in turn may improve the detection of periparturient metabolic disorders and other diseases. A possible approach with a Decision Tree model considering multiple sources of sensor data was proposed by Steensels et al. [91], combining rumination, activity, and milk yield to assess the probability of a cow being sick. The overall accuracy of the model was 78% and the sensitivity and specificity were 69% and 87%, respectively. Their results suggest that a post-calving health-detection model can be created using available sensors in a robotic-milking dairy farm; however, the use of additional data from additional sensors might improve the accuracy of the model. In a recent study, Zhou et al. [92] utilized eight machine learning algorithms to detect health issues in dairy cows, utilizing data from automated monitoring systems (AMS) and milking systems. The study emphasized the importance of using AMS data for predicting and monitoring health disorders in dairy cows, including variables such as milk yield, physical activity, changes in rumination time, and the electrical conductivity of milk.

### 4.2. Cow Behaviour and Hard Management

One potential benefit of using AMS is the ability to monitor cows’ individual patterns of physical activity, as cows in free-stall barns may develop unique patterns over time [93]. Despite the obvious benefits due to milking robots on the voluntary movement of cattle inside the barn, the daily human–cow interaction decreases with the change from a conventional to an AMS. This entails reduced direct supervision of the behavior and welfare of the animals from the farmer, and other ways of observing changes in cows’ behavior and welfare become necessary. Despite the limited time available to stockpersons for each animal, it remains crucial to actively monitor the behaviors of individual cows. Adamczyk et al. [94] classified the physical activity of dairy cows milked in the voluntary milking system using cluster analysis from data obtained by neck-mounted tags. Specifically, they classified the physical activity of cows by means of Ward’s method, a hierarchical agglomeration method, and Kohonen’s self-organizing map, a method based on NNs, with regard to varying environmental conditions. Physical activity during individual months showed small variability. However, over the individual months, the cluster obtained by Ward’s method highlighted different groups depending on daytime light length, temperature, and relative humidity. In this study, Kohonen networks were used only for the verification of Ward’s method and to test the similarity between clusters obtained with both methods. The rapid advancement of technology raises hope that in the future, the measurement of cows’ physical activity, particularly the most significant forms, will be even more precise. Furthermore, there is the potential for real-time analysis of this data [94]. The application of machine vision systems to recognize and monitor the activity and behavior of animals in a quantitative manner could become the solution needed [95]. Guzhva et al. [96] used top-view cameras to automatically detect social interactions (head pressing and body pushing). A two-step pattern recognition approach was used. First, the distances from every couple of cows were extracted. Then, an SVM was used to classify the behavior of cows. Guzhva et al. [97] implemented a tracking algorithm for cow detection and motion extraction, based on Convolutional Neural Networks (CNNs). The CNN-detector used was implemented in two steps:a full CNN that detects the landmarks in the image;a CNN that works with the probability map produced by the first CNN as input to detect the cows and their orientations.

Both studies [96,97] implemented a successfully non-invasive system capable of individual tracking and identification and the detection of social interactions. However, the region of interest for the recordings was limited to a waiting area with free entrances to AMSs (6 × 18 m). Considering the increasing average size of dairy herds and the number of individuals requiring monitoring, a computer vision system to track and monitor the social interactions and the space usage of the whole herd is required, and the potential to identify welfare-compromised animals through motion characteristics or spatial characteristics needed be explored. The great potential of the AMS datasets for herd characterization and management optimization is still underexploited [7]. Data of AMS are, for example, suitable to identify clusters within the herd with a focus to support the farm management in the herd segmentation decision. For this purpose, a K-means model was used to provide an automatic grouping of the cows based on production and behavioral features [7]. The time series data of cows milked in AMS were utilized to categorize herd characteristics and classify cows based on five distinct parameters: the number of daily milking procedures, parity, average daily activity, milking regularity, and cow body weight. K-means clustering models were employed for each of these parameters, enabling the characterization of the herd into clusters based on various productivity and behavioral features. The methodology, as proposed by the authors, was designed following general criteria that are not specific to a single case of application. Therefore, it can be applied to other study cases encompassing different herd characteristics.

### 4.3. Production

The AMS provides farmers with detailed data concerning parameters connected with milk production (such as milk yield, days in lactation, percentage of lactose, fat, and protein) which are of great interest to improve the farm performance in terms of milk quantity and quality. Farmers can benefit from accurate forecasting of milk yield to implement financial plans and to detect deviating yield patterns [98]. Dynamic linear modeling (DLM) was able to predict the cows’ individual milk yields per milking [98]. DLM offers several advantages, such as the handling of missing data because the forecast values are automatically adjusted over time considering the expected trend of the data [99]. This dynamic approach for estimating the expected milk yield per milking of individual cows was able to detect the deviation between observed and predicted milk production. Moreover, the DLM was affected by the SCC level, and a significant interaction between SCC and lactation stage was observed which suggests that the model could be used also to predict the cow’s health [98]. Decision Tree (DT) techniques have found several applications in predicting milk yield. Decision trees (DTs) offer several advantages, including their intuitive nature and the ease of interpreting the data presented as simple graphical models. These models allow for the analysis of the impact of individual factors as well as their interactions [100]. Piwczyński et al. [101] found that milking frequency, lactation number (parity number), the month of milking, and type of lying stall are significant factors influencing the monthly milk yield of dairy cows. At the same time, they demonstrated that there were several interactions between the aforementioned factors, the understanding of which is significantly facilitated by the DT techniques. Piwczyński et al. [101] demonstrated that the DT method, by analyzing the graphical model, allows herd managers to identify the factors that influence specific productive traits of animals. More recently, a Classification and Regression Trees (CART) Decision Tree algorithm was employed to predict lactation milk yield based on information recorded during the periparturient period [102]. CART is a ML technique that has been shown to be particularly valuable when analyzing nonlinear relationships and interactions, and identifying the variables that automatically affect and reduce the complexity of the data [103]. This study builds upon previous research by the authors, which focused on predicting the lactational milk yield of cows using data collected by AMS during the periparturient period [104]. In the earlier publication, the authors presented descriptive statistics related to the prediction of characteristics such as services per conception and calving intervals. The CART method showed that the most important factors responsible for lactation yield were the survival to the next calving, the milking time per visit and the number of milkings per day.

It is well-known that heat stress is an important factor that negatively influences lactating cows’ performance [105]. Recently, there have been implementations of machine learning (ML) modeling to analyze environmental factors, including the Temperature Humidity Index (THI), and its impact on heat stress in dairy cows. These applications aim to optimize the utilization of the abundant big data available from robotic dairy farms and understand its effects on the final productivity and quality of milk. A few studies used ML approaches to predict production traits in challenging climatic conditions, often evaluated using THI [18,106]. An RF algorithm was adopted to assess the trend in daily milk yield in relation to environmental conditions, both as a regression tool and a predictive tool in short and long periods [18]. The daily milk yield was assessed by considering the position of the day in the lactation curve and the daily average of the Temperature Humidity Index (THI) on that day, as well as its values on each of the five preceding days. The RF model detected the drop in the cow’s milk yield due to extremely hot conditions and represented a reliable tool for the evaluation of milk production in the presence of heat stress effects [18]. Fuentes et al. [106] used two ML models based on NN using the Bayesian regularization training algorithm. The first model used data from cows with similar heat tolerance, and the second one, data from all cows from the farm. The input data consisted of programmed concentrate feed and weight combined with microclimatic parameters, i.e., temperature, relative humidity, rainfall, wind speed, wind, and THI. Both models presented similar results with high accuracy to predict milk yield, milk fat, and protein content, and concentrate feed intake. A study conducted by Ji et al. [107] investigated the feasibility of utilizing data collected by AMS to forecast milk yield, milk composition, and milk frequency. The authors suggested various potential applications of their machine learning framework, including identifying cows that experience heat stress or health issues and providing accurate treatment, such as nutrient adjustment or cooling, using AMS data.

## 5. Conclusions and Future Directions

ML algorithms have emerged as widely used research tools in the livestock sector, offering valuable insights, especially in areas that require predictive capabilities. While traditional statistical methods have served as a fundamental basis for analysis, ML algorithms open up new avenues for more sophisticated data-driven discoveries. The aim of this review was to draw current knowledge on the use of modeling approaches to the data obtained specifically from the sensors of milking robots and, in general, to the data obtained from the farms equipped with AMS. We provided a literature review of 60 works with a specific focus on cows’ health, production and behavior/management from 2000 to 2022. Based on the selected studies, several interesting observations were determined. The most used modeling approach was ML (present in 63% of the studies), followed by statistical analysis (14%), fuzzy algorithms (9%), deterministic models (7%), and detection algorithms (7%). The number of articles increased slowly in the last few years, demonstrating a growing interest in the use of modeling approaches in the dairy sector, and in analyzing data from the AMS systems.

Most of the studies were intended for the detection of cows’ health problems (82%), notably subclinical and clinical mastitis. However, the application of ML techniques in this field still suffers from a lack of a robust methodology, which hinders the advancement of these studies. For example, in the reviewed studies whose goal was to model the mastitis occurrence, some used quarter-specific data, while others used the integrated data of all quarters to detect the disease. Although they deal with the same disease, from the modeling point of view, they are two distinct problems. Another issue identified in this review is the absence of a systematic balancing of the positive and negative classes for mastitis prediction models. As for mastitis data, the dominant class is the negative, unbalanced data lead to high specificity but low sensitivity, as observed in many studies, and makes the model comparison infeasible. However, the potentiality of ML approaches for the detection of other diseases, besides mastitis, from AMS data is still underexploited. Surprisingly, only 11% of the studies focused on milk production. Accurate forecasting of dairy cow milk yield and knowing the deviation between expected and observed milk yields of individual cows would be beneficial in dairy cow management. This raises the question of why ML methods are not being fully exploited to improve production strategies. One reason could be the lack of availability of multiparameter datasets that include more information on milk quality and quantity. Well-described, multifactorial and high-quality datasets would allow for the development of better algorithms for production management. Likewise, the study of cow behavior and herd management in AMS systems is under-explored. Since AMS relies on cows milking themselves voluntarily, in this system the cows are free to move and interact during the whole day and this allows the study of cows’ social interactions. Thus, farms equipped with milking robots represent a good environment to investigate physical activity and social networks through, for example, computer vision systems.

## Figures and Tables

**Figure 1 animals-13-01916-f001:**
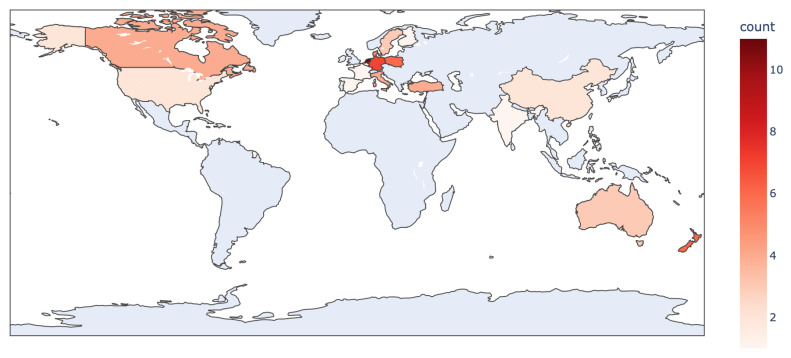
Geographical distribution of the 60 contributing studies in modeling approaches in farms equipped with AMS, considering the location of the dataset collection.

**Figure 2 animals-13-01916-f002:**
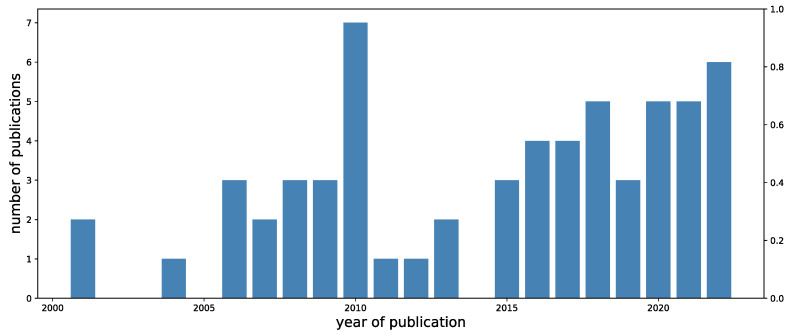
Timeline of number of publications included in this review from 2000 to 2022.

**Figure 3 animals-13-01916-f003:**
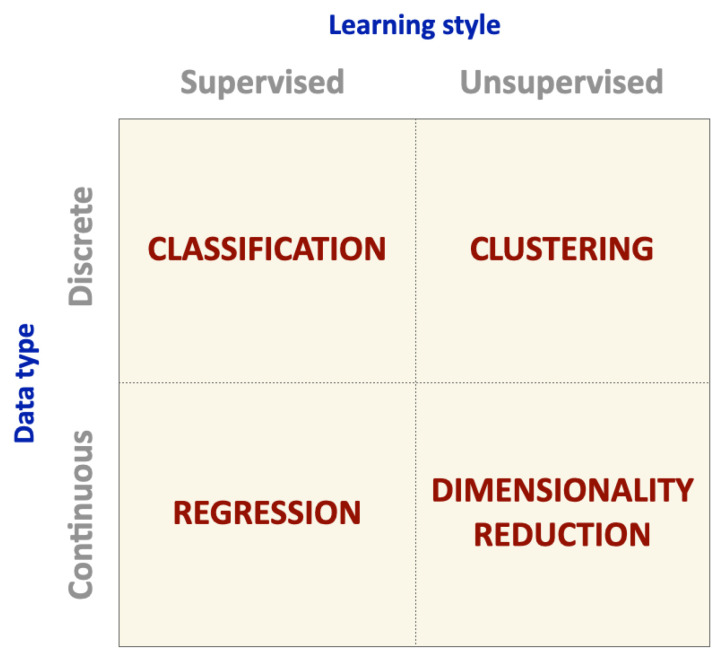
Types of supervised and unsupervised machine learning. Problems are grouped into four main classes: classification, clustering, regression, and dimensionality reduction based on data type (discrete or continuous) and learning style (supervised or unsupervised) (adapted from [10]).

**Figure 4 animals-13-01916-f004:**
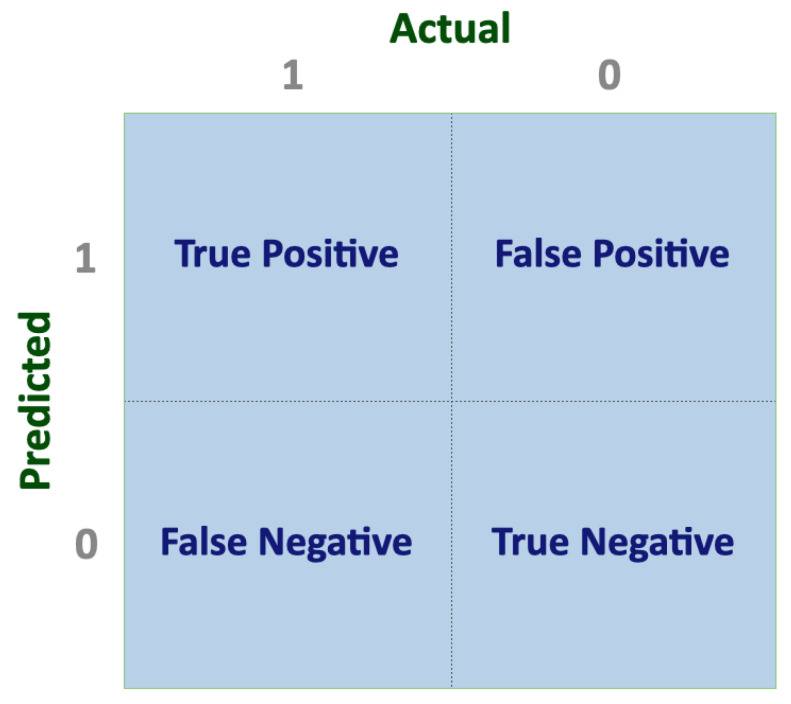
Representative illustration of a confusion matrix.

**Figure 5 animals-13-01916-f005:**
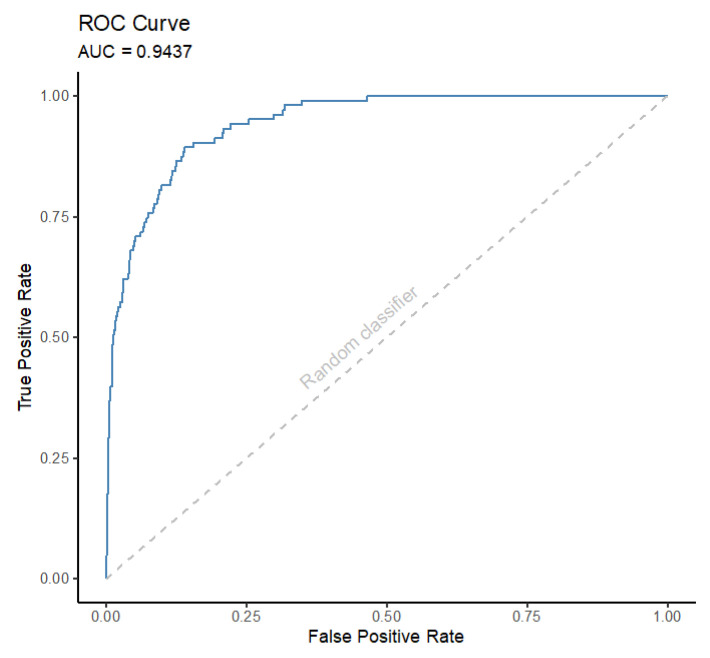
Example of a Receiver Operating Characteristic Curve (ROC).

**Table 1 animals-13-01916-t001:** Modeling techniques used in the studies reviewed (adapted from [24]).

Algorithm	Description
Regression analysis	Regression analysis is a statistical technique used to describe the relationships between variables. It allows predicting certain characteristics of output values based on input values [30]. It includes classical models such as simple and multiple linear regression, logistic regression, Generalized Linear Models (GLM), Generalized Additive Models (GAM), linear mixed models, polynomial regression, and time series.
Decision Tree (DT)	A decision tree (DT) is a predictor that associates the features values with a label of a data instance by traveling from a root node to a leaf of a tree structure. Each node represents the splitting of the input space [29]. The feature or the value to be used for this splitting depends on the problem. A common splitting rule is the maximization of the Information Gain, which is the reduction in information entropy on split groups. When used for regression problems, it is called CART, an acronym for Classification and Regression Trees.
Random Forest (RF)	Random Forest (RF) is an ensemble classification model that combines several randomized decision trees. These decision trees are models that classify random subsets of the data where each subset contains responses of one class (either “yes” or “no”) [31]. Additionally, different trees can also use different sets of features to be trained or different random subsets of the data. For the RF outcome, the decision trees predictions are combined in a disambiguation method, for example, averaging [32] in the case of regression problems or major
AdaBoost	AdaBoost stands for Adaptive Boosting. It is also referred to generically as Gradient Boosting. It combines sequentially the result of many weak decision trees. The first decision tree takes the raw data as input. The others receive as input the data weighted by the prediction errors of the previous classifier. Thus, each decision tree will adjust the prediction of the previous classifier.
k-Nearest Neighbors (k-NN)	The k-Nearest Neighbors (k-NN) algorithm examines labeled points in proximity to an unlabeled point, utilizing this information to predict the appropriate label [33]. Therefore, the learning strategy of k-NN is memorizing instead of finding relationships among features.
Support Vector Machine (SVM)	The objective of the Support Vector Machine (SVM) algorithm is to find the boundaries that maximize the distance of a multi-dimensional plane that separates the classes to be modeled. It uses the geometrical properties of the data to build these multi-dimensional boundaries between data points in the feature space belonging to different classes [34].
Bayesian Networks (BN)	Bayesian networks (BN) are a form of probabilistic graphical model that utilizes Bayesian inference for probability calculations. The primary objective of Bayesian networks is to capture conditional dependence and, consequently, causation by representing conditional dependence by edges in a directed graph [35]. The Naïve version assumes independence amongst the features, while the Tree-Augmented version also allows modeling the dependency amongst the features themselves.
Neural Networks (NN)	A Neural Network (NN) is a computational model that consists of three types of layers: an input layer, hidden layers, and an output layer. Each layer is comprised of nodes, also known as neurons. The outputs of each neuron are transformed using a nonlinear function and then passed to the subsequent layers through weighted connections between neurons. The input layer receives numerical data representation, while the output layer generates predictions. The hidden layers carry out nonlinear transformations on the data [36]. Various types of NN include Multilayer Perceptron (MLP), Back Propagation Neural Network (BPNN or NN for short, since this is the most used neural network), Probabilistic Neural Network (PNN), Recurrent Neural Network (RNN), Convolutional Neural Network (CNN). A CNN is a Deep Learning algorithm, which means that its network architecture usually needs a high number of hidden layers. The CNN takes in an input image, and, in these hidden layers, it assigns importance (learnable weights and biases) to various aspects/objects in the image. Then, this transformed image is used for the classification in the output layer.
Self-Organizing Maps (SOM)	Self-Organizing Maps (SOM) are a different type of Neural Network. They show only one layer with a predefined number of nodes. These nodes are linked to the input data and the value associated with these connections represents the distance between them. Thus, the SOM can be seen as a two-dimensional representation of the data, in which the data structure is preserved. The most common use of this algorithm is in clustering analysis, which requires a post-processing phase in which the SOM nodes will be clustered.
Clustering Algorithms	Clustering algorithms are designed to partition objects into groups, known as clusters, based on measures of similarity and dissimilarity among the objects. The goal is to maximize the similarity among objects of the same cluster and, at the same time, maximize the dissimilarity among objects that belong to different clusters. Examples of these measures include one minus correlation and Euclidean distance. Two commonly used clustering techniques are hierarchical clustering (HC) and k-means clustering. Hierarchical clustering creates a hierarchical tree-like structure, where the length of the branches represents the dissimilarity between clusters. The hierarchical tree is cut at some point and the branches that are separated at this cut will define the clusters of the objects.The k-means clustering algorithm starts with random cluster centers (k), the number of these clusters is specified by the user [26]. The data points are assigned to the nearest cluster center. The center clusters are, then, redefined according to the new cluster configuration. This process is repeated iteratively until the cluster centers are no longer modified or until a maximum number of iterations.
Fuzzy logic	In Fuzzy Logic theory, the objects do not belong exclusively to one set (or class) or to another. Instead, they have a continuum of grades of membership to all classes, varying from 0 to 1 [37]. Fuzzy logic-based decision support systems usually follow three basic steps. First, the input values are fuzzified by the assignment of the membership functions. Second, a set of logic rules are applied to transform the input values, generating the output. Lastly, these outputs are defuzzified to generate the crisp system prediction. It is a method specifically designed to handle situations where there are highly non-linear relationships between input and output variables, aiming to achieve the optimal solution. As a special case, the Adaptive Neuro Fuzzy Inference System (ANFIS) is a NN to map numerical inputs into an output through fuzzy-based rules.
Genetic Algorithms (GA)	Genetic Algorithms (GA) are search algorithms of the family of Evolutionary Algorithms based on the mechanics of natural selection and natural genetics [38]. The individuals are the possible solutions for the problem to be optimized. The set of these individuals that evolve together form the algorithm’s population, and the fitness of the individuals is the criteria for a probabilistic selection of the solutions, in which the better the fitness, the higher the probability of that individual being selected for the next generation. This type of stochastic search algorithm is often used in ML applications [39]. GAs are used in discrete spaces and find applications in cases where other gradient-based methods are not applicable. GAs are well-suited for situations where the availability of information plays a crucial role in performance [24].

**Table 2 animals-13-01916-t002:** Summary of the most used performance metrics for classification tasks [29].

Performance Metric	Formula	Description
Accuracy	$\frac{TP\;+\;TN}{TP\;+\;TN\;+\;FP\;+\;FN}$	It is the ratio between the number of correct predictions versus the total number of input samples.
Error rate	$\frac{FP\;+\;FN}{TP\;+\;TN\;+\;FP\;+\;FN}$	It is the ratio between the number of wrong predictions versus the total number of input samples.
Sensitivity (Recall)	$\frac{TP}{TP\;+\;FN}$	It measures the proportion of correctly identified positive values.
Specificity	$\frac{TN}{TN\;+\;FP}$	It measures the proportion of correctly identified negative values.
Precision	$\frac{TP}{TP\;+\;FP}$	It is the proportion of positive predictions that are correct.
F1 score	$2\times \frac{\left(precision\;\times \;recall\right)}{T(precision\;+\;recall)}$	It combines precision and sensitivity in a harmonic mean.
Matthews correlation coefficient (MCC)	$\frac{TP\;\times \;TN\;+\;FP\;\times \;FN}{\sqrt{\left(TP\;+\;FP\right)\times \left(TP\;+\;FN\right)\times \left(TN\;+\;FP\right)\times (TN\;+\;FN)}}$	It considers all four values in the confusion matrix, and a high value (close to 1) means that both classes are predicted well

## Data Availability

Not applicable.

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
