# Peer review of "A Literature Review of Modeling Approaches Applied to Data Collected in Automatic Milking Systems"

_animals, 2023, doi:10.3390/ani13121916_

Round 1
Reviewer 1 Report
This is an interesting and well written review paper. The matter is of increasing interest for the research community and the properly and accurate use of data from milking systems is an open issue in the research field paving the way for future improvements of the tools made available to the farmers. The topic falls within the scope of the Journal. Therefore, I recommend the paper for publication after minor revisions addressing the folloewing comments:
1) The paper focuses on the description and application of data collected by AMS on dairy cows. This aspect should be more clearly highlighted also in the title.
2) The paper needs a check of the English language with typo correction included.
Author Response
#Reviewer 1
This is an interesting and well written review paper. The matter is of increasing interest for the research community and the properly and accurate use of data from milking systems is an open issue in the research field paving the way for future improvements of the tools made available to the farmers. The topic falls within the scope of the Journal. Therefore, I recommend the paper for publication after minor revisions addressing the following comments:
1) The paper focuses on the description and application of data collected by AMS on dairy cows. This aspect should be more clearly highlighted also in the title.
Response: Many thanks for the suggestion. Accordingly, we have revised the title of the manuscript as follows: “A Literature Review of Modeling Approaches Applied to Data Collected in Automatic Milking Systems”
2) The paper needs a check of the English language with typo correction included.
Response: Thank you for the suggestion. We have revised the English language and corrected any typos.

Reviewer 2 Report
The authors provided a comprehensive literature review on the use of different modeling approaches applied to data retrieved from automatic milking systems. Overall, the manuscript is well written and the topic is of interest for the scientific community.
Please, consider the following comments to improve the paper.
L. 153-156. I would not say that “few” training examples are sufficient for accurate predictions. The algorithm needs to learn, so an adequate training set is necessary. Indeed, depending of the sample size, 75-80% of the data are usually utilized for the training phase and 20-25% for the testing.
L. 164. The partitioning can be random, but also stratified (for instance if in classification problems we want to preserve the proportion of data in each class of the outcome).
L. 164-169. The input data are usually divided into multiple data sets: training, validation, and test sets. At first, data are split into 2: training and testing sets (e.g., 80% and 20% of data). The testing set (20%) is kept aside, whereas X% of the training set (80% of original data) is chosen to be the actual training set and the remaining (100-X)% to be the validation set. K-fold cross validation allows to generate multiple splits of the training/validation sets. In the manuscript, it is not very clear the difference between testing and validation sets. I might suggest also to use a figure to better explain the split (see for instance https://learningds.org/ch/16/ms_cv.html).
Table 2. I would add also the Matthews correlation coefficient.
L. 206-209. What about the evaluation and comparison of regression models?
L. 440. Replace LHD with LDH
L. 514-516. In this study, test-day milk samples were collected during routine milk recording procedures…so I guess these are not data from AMS but from traditional milking parlor.
Author Response
#Reviewer 2
The authors provided a comprehensive literature review on the use of different modeling approaches applied to data retrieved from automatic milking systems. Overall, the manuscript is well written and the topic is of interest for the scientific community.
Please, consider the following comments to improve the paper.
153-156. I would not say that “few” training examples are sufficient for accurate predictions. The algorithm needs to learn, so an adequate training set is necessary. Indeed, depending of the sample size, 75-80% of the data are usually utilized for the training phase and 20-25% for the testing.
Response: Thank you for the valuable suggestion and explanation. We completely agree. However, as advised by other reviewers, we have removed this sentence from the text.
164. The partitioning can be random, but also stratified (for instance if in classification problems we want to preserve the proportion of data in each class of the outcome).
164-169. The input data are usually divided into multiple data sets: training, validation, and test sets. At first, data are split into 2: training and testing sets (e.g., 80% and 20% of data). The testing set (20%) is kept aside, whereas X% of the training set (80% of original data) is chosen to be the actual training set and the remaining (100-X)% to be the validation set. K-fold cross validation allows to generate multiple splits of the training/validation sets. In the manuscript, it is not very clear the difference between testing and validation sets. I might suggest also to use a figure to better explain the split (see for instance https://learningds.org/ch/16/ms_cv.html).
Response: We express our sincere gratitude to the reviewer for their valuable and highly regarded comments, which we find immensely helpful. However, considering the feedback from other reviewers, we have decided to remove this section from the review as it was deemed excessively long and detailed for its intended purpose.
Table 2. I would add also the Matthews correlation coefficient.
Response: Many thanks for the suggestion. Accordingly, we have added the Matthews correlation coefficient to the table.
206-209. What about the evaluation and comparison of regression models?
Response: Thank you for the suggestion. We greatly appreciate it. We have addressed the evaluation and comparison of the regression models in the dedicated section titled "Application of Modeling Approaches in AMS". Additionally, we have provided a detailed description of the regression models in Table 2.
440. Replace LHD with LDH
Response: Thank you, we have corrected the typo.
514-516. In this study, test-day milk samples were collected during routine milk recording procedures…so I guess these are not data from AMS but from traditional milking parlor.
Response: We have revised the paper once again and found that the data is sourced from both milking parlors and AMS.

Reviewer 3 Report
Line 14: "don't always balance positive and negative cases" is not entirely clear, can you clarify by adding "the number of positive and negative cases is often unequal, leading to populations that are not balanced.
Line 30-31: the last sentence of the abstract is not clear, please modify
Line 39-40: some of the nuance is lost with free vs. guided flow systems and animals are not always free to visit the milking parlor at any time.
Line 42: there is not consensus in the literature that AMS reduces labor, it may redirect labor or reduce labor associated with milk harvest
Line 44: individual cows may be milked more than 3 times per day
Line 58: managed differently than traditional milking systems?
Line 72: already defined machine learning, throughout, be aware of what you have defined already and no need to define again
Line 89-92: please consider rewriting this sentence as it is hard to read in its current form
Line 109: unclear what you mean by these articles also met the search scope, does that mean that they were already found in your search?
Line 137: Description of figure likely needs to be more descriptive so it can stand alone.
Line 156-169: this reads as repetitive and in much more depth than the next two methods.
Line 170-173: can this be expanded to include how these models are assessed, it reads as much less developed than the supervised learning section.
Line 175-179: I am not sure if you need this part as you do not bring it back up and it is underdeveloped at this point
In Table 1: k-NN, I don't think an algorithm looks is there a different word choice?
Table 1: SOM, "the" instead of "de"
Table 1: Fuzzy logic, if there is a way to not use the word fuzzy as much in the description, it may help the reader
Line 217-221: change examples to values
Line 274-275: this sentence is not clear, please consider revising
Line 317-322: were all studies measuring color of milk the same way?
Line 333: you use both days from calving and days in milk, I think DIM would be ok throughout
Line 365-366: not clear what is meant here, it is ok to have a shorter training model or something different?
Line 396: SCC history - is that lifetime or this lactation?
Line 404: does 70% sensitivity classify as a successful method?
Line 462: what is classified as an udder treatment? antibiotics or something different?
Line 522: Does that mean only 6 were used for the testing data set?
Line 537 and 539: unclear what you mean by major and minor pathogens as well as small and large pathogens here, please clarify
Line 610: how are behaviors ensured? I am not clear what that means.
Line 631: identify the cow within the image?
Line 645-652: I am having difficulty following this section.
Minor suggestions for wording:
Line 7: "are changing the game" is informal consider revising
Line 10: We focused on cows' health.... I believe you focused the literature review on published papers that focused on cows' health....
Simple summary: if possible reduce the number of times "we" is used in scientific writing
Line 18: AMS is defined as automatic milking systems, you don't need the s after AMS
Line 25: please consider "evaluated" rather than "were concerned with"
Line 85: AMS includes system you don't need to state system
Line 254: change "udder infections" to "mastitis"
Line 431: "viewed as dichotomous (healthy vs. sick)"
Line 729-730: word choice for "which hinder the development of these studies" do you mean the adoption of these techniques or something different? The studies have been completed, do you mean other studies using these techniques?
Line 731: "quarter-specific"
Author Response
#Reviewer 3
Line 14: "don't always balance positive and negative cases" is not entirely clear, can you clarify by adding "the number of positive and negative cases is often unequal, leading to populations that are not balanced.
Response: Thank you for the suggestion. To clarify this point we have rephrased the sentence as recommended by the reviewer: “However, there is still a lack of a robust methodology for utilizing ML techniques in this domain, and it was also observed that the number of positive and negative cases is often unequal, leading to populations that are not balanced when predicting health issues.”
Line 30-31: the last sentence of the abstract is not clear, please modify
Response: As suggested, we rephrased the sentence as follows: “Specifically, we found a substantial disparity in adequately balancing the positive and negative classes within health prediction models.”
Line 39-40: some of the nuance is lost with free vs. guided flow systems and animals are not always free to visit the milking parlor at any time.
Response: We agreed with the reviewer comment. In these two lines, our intention was to emphasize the contrasting approach to cow milking between AMS (voluntary visits) and traditional milking parlors (supervised by human handlers).
Line 42: there is not consensus in the literature that AMS reduces labor, it may redirect labor or reduce labor associated with milk harvest
Response: We totally agreed with the reviewer comment. Thus, we have modified the sentence as follows: “The AMS process is entirely mechanized, reducing the labor burden on farmers in relation to milking operations. This has the potential to enhance the quality of their work and improve their overall lifestyle.”
Line 44: individual cows may be milked more than 3 times per day
Response: It is possible that the cows had milked more than 3 times, but this is the average, thus we added the words “in average” in the sentence.
Line 58: managed differently than traditional milking systems?
Response: We have taken the reviewer's comment into account, and to provide further clarity, we have revised the sentence as follows: “The implementation of AMS technology not only facilitates the collection of milk quantity and quality data but also creates an opportunity to study cow behavior and welfare within a system that allows free cow traffic.”
Line 72: already defined machine learning, throughout, be aware of what you have defined already and no need to define again
Response: Accordingly, we have removed the repetition.
Line 89-92: please consider rewriting this sentence as it is hard to read in its current form
Response: According to the reviewer’s comment, we have we have rephrased the sentence as follows: “Motivated by the rapid advancements of ML, its increasing global popularity, and its potential influence on PLF (Precision Livestock Farming), we conducted a literature review focusing on various modeling approaches, including ML”
Line 109: unclear what you mean by these articles also met the search scope, does that mean that they were already found in your search?
Response: The articles included in this review were primarily sourced from the Web of Science and Google Scholar platforms. To identify relevant scientific articles, we utilized keywords encompassing various categories, such as "Machine Learning," "Artificial Intelligence," "modeling approaches," "Automatic Milking Systems" (or "milking robots"), and "Precision Livestock Farming" (or "Precision Dairy Farming"), including both the abbreviated and full names. We have clarified this point in the text.
Line 137: Description of figure likely needs to be more descriptive so it can stand alone.
Response: Accordingly, we added more information in the caption of the figure.
Line 156-169: this reads as repetitive and in much more depth than the next two methods.
Line 170-173: can this be expanded to include how these models are assessed, it reads as much less developed than the supervised learning section.
Response: We completely agree with the reviewer's comments, and as advised, we have shortened the explanation of supervised learning in comparison to the other two methods. Nevertheless, we opted to provide more extensive information on supervised learning due to its prevalence as the most commonly employed method in the articles included in the review.
Line 175-179: I am not sure if you need this part as you do not bring it back up and it is underdeveloped at this point
Response: Accordingly, we have removed this part.
In Table 1: k-NN, I don't think an algorithm looks, is there a different word choice?
Response: According to the reviewer’s comment, we have rephrased the sentence as follows: “The k-Nearest Neighbors (k-NN) algorithm examines labeled points in proximity to an unla-beled point, utilizing this information to predict the appropriate label.”
Table 1: SOM, "the" instead of "de"
Response: Thank you, we have corrected the typo.
Table 1: Fuzzy logic, if there is a way to not use the word fuzzy as much in the description, it may help the reader
Response: Accordingly, we have reduced the word “fuzzy” where possible.
Line 217-221: change examples to values
Response: Accordingly, we have substituted the word “examples” with “values”
Line 274-275: this sentence is not clear, please consider revising
Response: Accordingly, we have revised the sentence as follows: “Mastitis is a highly concerning condition that results in decreased milk production, reduced milk quality, and compromised cow welfare.”
Line 317-322: were all studies measuring color of milk the same way?
Response: Yes, all studies used the same color sensors.
Line 333: you use both days from calving and days in milk, I think DIM would be ok throughout
Response: Thank you for the comment, however, we preferred to keep the same wording as presented in the original cited paper.
Line 365-366: not clear what is meant here, it is ok to have a shorter training model or something different?
Response: Accordingly, to clarify this point we have rephrased the sentence as follows: “However, altering the input requirement from 30 days to 15 days had a negligible impact on the performance of the model.”
Line 396: SCC history - is that lifetime or this lactation?
Response: Here, when referring to "SCC history," we intended to convey the cumulative SCC levels throughout the cow's lifetime, and we have clarified this explicitly in the text.
Line 404: does 70% sensitivity classify as a successful method?
Response: We totally agreed with the reviewer’s comment and we have removed the term “successful”.
Line 462: what is classified as an udder treatment? antibiotics or something different?
Response: The mentioned “udder treatments” are antibiotics, including intramammary and systemics antibiotics.
Line 522: Does that mean only 6 were used for the testing data set?
Response: Yes, this is the data provided in the article.
Line 537 and 539: unclear what you mean by major and minor pathogens as well as small and large pathogens here, please clarify
Response: Accordingly, to clarify this point we have modified the sentence as follows: Bacteria that cause mastitis can be grouped into contagious or environmental, gram-positive, or gram-negative, or major and minor pathogens, according to their prevalence and the severity of symptoms.
Line 610: how are behaviors ensured? I am not clear what that means.
Response: Accordingly, to clarify this point we have modified the sentence as follows: “Despite the limited time available to stockpersons for each animal, it remains crucial to actively monitor the behaviors of individual cows.”
Line 631: identify the cow within the image?
Response: Yes, these techniques allow the identification of the cows within the images.
Line 645-652: I am having difficulty following this section.
Response: Accordingly, we have rephrased the sentences to enhance the readability of the section.
Comments on the Quality of English Language
Minor suggestions for wording:
Line 7: "are changing the game" is informal consider revising
Response: Accordingly, we have rephrased the sentence as follows: “Automatic milking systems (AMSs) are revolutionizing dairy farming worldwide.”
Line 10: We focused on cows' health.... I believe you focused the literature review on published papers that focused on cows' health....
Response: Thank you for the suggestion, we have rephrased the sentence as follows: “Our review primarily encompassed published articles addressing cows' health, production, and behavior/management. Within this field, Machine Learning (ML) emerged as the prevailing modeling approach.”
Simple summary: if possible reduce the number of times "we" is used in scientific writing
Response: Thank you for the suggestion, we have rephrased some sentences in the simple summary with the aim to reduce the number of “we”.
Line 18: AMS is defined as automatic milking systems, you don't need the s after AMS
Response: Accordingly, we have removed the “s” as recommended by the reviewer.
Line 25: please consider "evaluated" rather than "were concerned with"
Response: Accordingly, we have replaced the terms "were concerned with" with "evaluated".
Line 85: AMS includes system you don't need to state system
Response: Thank you, we have removed the term “system”.
Line 254: change "udder infections" to "mastitis"
Response: Accordingly, we have replaced the term "udder infections" with "mastitis".
Line 431: "viewed as dichotomous (healthy vs. sick)"
Response: Thank you for the suggestion. We have made the necessary correction to the sentence as recommended by the reviewer.
Line 729-730: word choice for "which hinder the development of these studies" do you mean the adoption of these techniques or something different? The studies have been completed, do you mean other studies using these techniques?
Response: To provide further clarity, we have revised the sentence as follows: "However, the application of ML techniques in this field still suffers from a lack of a robust methodology, which hinders the advancement of these studies."
Line 731: "quarter-specific"
Response: We have rectified the error in the term "quarted-specific" and replaced it with the correct term "quarter-specific". We sincerely apologize for the typo mistake.

Round 2
Reviewer 3 Report
Thank you for the prompt responses and revisions.